# Performance Evaluation of Multiflight Ground Handling Process

**Biao Li** [1] , **Liwen Wang** [1], **Zhiwei Xing** [2,*] and **Qian Luo** [3]

1   College of Aeronautical Engineering, Civil Aviation University of China, Tianjin 300300, China;
    bill.lee.cauc@gmail.com (B.L.); cauc_wlw@126.com (L.W.)
2   College of Electronic Information and Automation, Civil Aviation University of China, Tianjin 300300, China
3   Engineering Technology Research Center, The Second Research Institute of Civil Aviation Administration of
    China, Chengdu 610041, China; caacsri_luoqian@163.com
*   Correspondence: cauc_xzw@163.com

**Abstract:** The ever-increasing high density of flights arouses an urgent requirement to improve the effectiveness and performance of ground handling in airport operation. The implementation of coordinated airport decision-making quantifies the ground handling process into a series of key milestone nodes, which is more conducive for operators to reduce resource consumption and flight delays. An innovative performance evaluation method for the multiflight ground handling process is proposed based on shared information of milestone nodes in the ground handling. A dynamic performance evaluation model is established, which should superimpose the performance evaluation results of the single-flight ground handling process. Meanwhile, the indicators and weights of the single-flight performance evaluation are obtained by combining the ground handling process prediction and expected value. As time evolves, a matrix method for the multiflight ground handling performance evaluation is proposed to combine the logic and evolution of the process. It is shown that the average prediction accuracy of single-flight ground handling process nodes can be increased to 87.63%. The experimental analysis demonstrates that the objectivity, effectiveness and dynamics of the proposed approach can be the basis for short-term tactics in airport.

**Keywords:** air transportation; performance evaluation; node prediction; matrix transformation; ground handling process





## 1. Introduction

The flight ground handling process (FGHP) is an important part of milestone events for airport collaborative decision-making (A-CDM) implementation to realize the next generation's smart airport system [1]. FGHP determines whether the flight can be launched according to the normal flight schedule, which may have a propagation effect, resulting in a decline in the overall operating efficiency of the airport. At present, FGHP is jointly conducted by airports, airlines and ground service companies, and the evolution of FGHP is constrained by the prediction of the operation situation based on the real-time shared data from the airport operation control center (AOCC) [2]. However, the dynamic FGHP data for evaluating the multiflight performance can provide AOCC with an objective and effective decision-making basis [3]. Thus, the impact of ground handling process uncertainty and correlation on airport operations may be basically eliminated.

Flight-oriented performance evaluation methods have been proposed in [4,5] to guide the long-term tactical formulation and improve safety of airports. These methods evaluate performance and safety in different forms. However, the ground handling resources involved in FGHP are of many types and varieties, and the process series-parallel hybrid evolution is not systematic, synergistic and stable, which causes the node dynamic operation situation of FGHP to be regarded as nondeterministic polynomial hard problem (NP-Hard). Because the essence of FGHP is NP-Hard [6], such an evaluation method does

not achieve real-time evaluation of the operating situation and cannot be supported in a dynamic airport operation environment [7]. To obtain the future performance and operating situation of FGHP, more approaches based on FGHP prediction have been studied using a stochastic process model to analyze the correlation of each subprocess, which can basically infer the inherent logic. A real-time monitoring prototype system was constructed [8] using the Markov probability statistical simulation [9] and Monte Carlo method [10] to simulate the disturbance caused by an uncertain event. The FGHP time node historical database was established; it can design a probabilistic learning model based on aircraft space–time position data from the perspective of tactical organization [11]. Based on historical samples, effectiveness evaluation was explored by considering the difference between actual operations and historical averages, which still cannot affect short-term airport tactical decisions [12]. Therefore, the process prediction and evaluation methods were studied based on linear regression and distribution fitting, considering the coupling between multiple flights; the unsupervised classification learning based on the Gaussian mixture model was implemented to realize the cooperative decision-making of airport operations [13,14], but multimodal characteristics lead to low prediction accuracy. A multiconstrained mathematical model was established with the core of passenger service to find a feasible performance evaluation method, and a flight ground handling agent system [15] was constructed from the safety regulations. Strong, transparent and independent safety performance evaluation methods ensure the effective management of the aviation management system, which is widely applied by hub airports around the world. In addition, an airport operation safety assessment system has been constructed from a macro perspective, including tools such as hazard identification, fault tree analysis and safety data recording [16].

Generally, when the airport operation risk performance is rather elementary, the number of FGHPs should be increased to improve capacity [17] and punctuality [18]. Describing the same node of FGHP modularly and exploring the interrelationship intensively between the various links of FGHP nodes are necessary. The airport departure saturation applied to characterize the comprehensive performance evaluation results of a multi-FGHP can provide a certain basis for the flight pushback control strategy, but it cannot play a decisive role in the whole airport performance [19]. Meanwhile, the management of departure flight queues based on Markov stochastic process is a systematic methodology [20] to characterize airport performance in capacity; it can also be regarded as a foundation of airport operation control decision support tools. A random runway capacity performance evaluation model considering random elements was established, and an airspace system capacity manual was developed based on relevant actual operating standards and rules [21,22]. The airport operation shared data were obtained, and the performance indicators are analyzed based on the linear regression method. Thus, the capacity performance evaluation model [23] was constructed and revised in accordance with the actual configuration situation. Consequently, the airport performance indicators and capacity analysis framework were developed. Furthermore, FGHP's improved Petri net model was promoted to carry out unsupervised learning, and the neural network method was used to start from the airport operation perception mechanism while ignoring that FGHP is a process of dynamic evolution [24,25]. In addition, the probability learning model based on aircraft space–time position data was designed to implement the Gaussian mixture model. However, the convergence of the model is insufficient, and the performance evaluation of FGHP lacks an argument to support the conception of A-CDM [26]. To sum up, the theoretical system for evaluation of airport operation safety and risk is relatively complete and can provide support for airport accident prevention and long-term operation strategies. However, it cannot guide the cooperative operation analysis and decision-making of short-term sudden changes. In view of this gap, a performance evaluation based on the dynamic evolution regularity of FGHP urgently needs to be constructed to meet the demand for autonomous airport operation.

The authors propose a dynamic performance evaluation method for multi-FGHP-based shared data from AOCC to perceive the situation of airport operation handling and

ground support resources. FGHP is described as a topological network with an association relationship according to each subprocess, and FGHP nodes can be predicted by a Bayesian network based on the dynamic evolution of the process. Combining the predicted results and the expected results from historical data statistics, dynamic performance indicators and weights are calculated to update the value of a single FGHP performance evaluation. A multi-indicator coupled FGHP performance evaluation method is designed based on the correlation matrix, and the multi-FGHP performance evaluation method is manifested to illustrate the airside situation. The multi-FGHP performance evaluation result can ensure that the airport operation decision-makers understand the overall status of the massive operation data.

The remainder of this paper is organized as follows: In Section 2, we systematically introduce the corresponding principles and constraints of FGHP and construct the dynamic time prediction theory of flight ground handling nodes based on a dynamic Bayesian network. Section 3 presents the innovative performance evaluation methods of multi-FGHP based on one flight unit. In addition, a case study is presented in Section 4 to prove the objectivity and rationality of the proposed approach. Section 5 concludes the study and puts forward the future research directions.

## 2. Materials and Methods

### 2.1. Frame of FGHP

Ground handling is important in guaranteeing the punctuality rate of flights, including all the aircraft ramp activities in the contact and remote stands. Generally, FGHP involves four parallel subprocesses, namely refueling, cargo bulk unloading/loading, maintenance inspection and passenger-cabin-related and other servicing. Passenger-cabin-related servicing also includes parallel flows, such as potable water servicing, toilet servicing, catering and cleaning.

The typical FGHP space layout [27,28] is shown in Figure 1, and the entire FGHP is strictly required to be executed in accordance with airport operation procedures and standards, which may be derived from the IATA AHM 810 Standard Ground Handling Agreement [29]. The figure shows a certain causal relationship between the various service links, and it is constrained by ground support equipment configuration, scheduling path and time and sequencing of flight arrival and departure queues. At the same time, being constrained by aircraft type, boarding stand, peak hours and airline [30], the network spread of the entire airport delay is very likely to advance or lag along with a guarantee node of a certain flight. In summary, the FGHP is a multimodal process problem with multiple support resource coordination, strict time window restrictions and complex condition constraints.

According to the actual process and relevant standards [31], each key node is designed and idealized as follows:

i.   Certain synchronization and short-term continuity exist in the three operations of on/off block, bridge docking/withdrawal and opening/closing cabin (cargo) door. Thus, they are regarded as one node.

ii.  The coupling effect is not considered, and the propagation effect of the guarantee resource allocation and scheduling is ignored generally because the initial moment of each ground handling node is greatly affected by external factors and cannot directly reflect the evolution of the entire process.

iii. The adjustment and correction of the FGHP of the dynamic queue sequencing of the ATC tower flight arrival/departure are not considered.

The network topology of the FGHP after being idealized is presented in Figure 2. As the basic logic of FGHP, a time and space sequence exists between the nodes. The balance and stability of the entire process are determined by the quality of each node. Thus, each node of each flight affects the effectiveness of airport operation. Therefore, the multi-FGHP performance evaluation method is constrained by the evolution and prediction of a single FGHP for decision making in airport cooperative operation control.

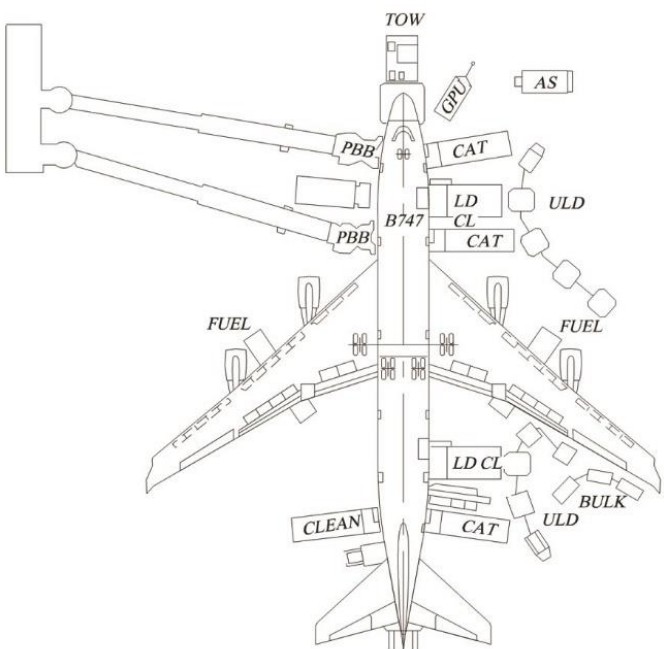

**Figure 1.** Typical FGHP space layout. TOW: tow tractor; GPU: ground power unit; AS: air start unit; PBB: passenger boarding bridge; CAT: catering; LDCL: cargo loading; ULD: unit load devices; FUEL: refueling; CLEAN: cleaning; BULK: bulk dollies.

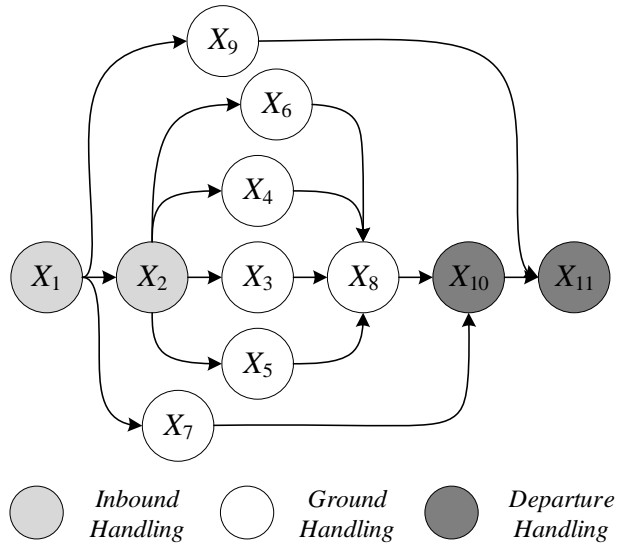

**Figure 2.** Idealized network topology graph of FGHP. $X_1$: on block/docking bridge (stairs)/open cabin (cargo) doors. $X_2$: passenger deplaning ending. $X_3$: cleaning ending. $X_4$: garbage truck operation completed. $X_5$: catering completed. $X_6$: refueling. $X_7$: uploading cargo manifest. $X_8$: passenger permitting. $X_9$: maintenance inspection confirmation. $X_{10}$: passenger boarding ending. $X_{11}$: undocking bridge (stairs)/close cabin (cargo) door/off block.

## 2.2. Time Prediction of FGHP Node

To obtain a comprehensive performance evaluation for a single FGHP, it is necessary to perceive the evolution of FGHP by node prediction. Therefore, the discrete-time topology network of FGHP (shown in Figure 2) is regarded as a dynamic Bayesian network model with a fixed structure and evolution over time [32,33], where $X_1$ is the root node, $X_{11}$ is the leaf node and loopback is excluded.

For the child node $X_i$ of $X_1$, a conditional probability model is as follows:

$$f(X_i|X_1) = \frac{f(X_1, X_i)}{f(X_1)} \quad i = 2, 7, 9 \tag{1}$$

where $f(X_1, X_i)$ is the joint probability model of the occurrence of node $X_1$ and child node $X_i$ simultaneously, and $f(X_1)$ represents the prior probability model of the occurrence of node $X_1$.

At $X_i$ with $X_2$ as its parent node, a conditional probability model developed as follows:

$$f(X_i|X_1, X_2) = \frac{f(X_1, X_2, X_i)}{f(X_1, X_2)} \; i = 3, 4, 5, 6 \tag{2}$$

where $f(X_1, X_i)$ is the joint probability model of $X_1$, $X_2$ and $X_i$. $f(X_1, X_2, X_i)$ refers to a prior probability model of the occurrence of nodes $X_1$ and $X_2$.

Towards $X_1$, the child node $X_8$ common to $X_3$, $X_4$, $X_5$ and $X_6$, a conditional probability model is calculated as follows:

$$f(X_8|X_1, \ldots, X_6) = \frac{f(X_1, \ldots, X_6, X_8)}{f(X_1, \ldots, X_6)} \tag{3}$$

where $f(X_1, \ldots, X_6, X_8)$ is the joint probability model of nodes $X_1$ to $X_8$ (except $X_7$), and $f(X_1, \ldots, X_6)$ implies a prior probability model of nodes $X_1$ to $X_6$.

Similarly, we can deduce that

$$f(X_{10}|X_1, X_2, \ldots, X_8) = \frac{f(X_1, \ldots, X_8, X_{10})}{f(X_1, X_2, \ldots, X_8)} \tag{4}$$

$$f(X_{11}|X_1, X_2, \ldots, X_{10}) = \frac{f(X_1, X_2, \ldots, X_{11})}{f(X_1, X_2, \ldots, X_{10})} \tag{5}$$

where $f(X_1, \ldots, X_8, X_{10})$ is the joint probability model that other FGHPs, except node $X_9$, proceed contemporaneously, and $f(X_1, X_2, \ldots, X_8)$ is the a priori probability model, where all ancestor nodes of $X_{10}$ occur. $f(X_1, X_2, \ldots, X_{11})$ is the joint probability model, where all nodes of FGHP occur synchronously [34], and $f(X_1, X_2, \ldots, X_{10})$ represents an a priori probability model, where all nodes in the Bayesian network of FGHP are achieved except the leaf nodes. According to the evolution and distribution of FGHP, the idealized chain rule indicates the following:

$$f(X_1, \ldots, X_m) = f(X_1)f(X_2|X_1) \ldots f(X_m|X_1, \ldots, X_{m-1}) \tag{6}$$

where node $X_m(1 < m < 11)$ is the parent node of node $X_m(m < i < 11)$, and the Bayesian network probabilistic inference model of event-based FGHP is acquired. The passenger service process has parallel and series flows [35]. Thus, the FGHP Bayesian network is inferred dynamically to ensure that the network can be parsed [36].

**Theorem 1.** *If $k(k \geq 1)$ parent nodes with node $X_i$ exists, then the prior probability model and joint probability model are interpreted as follows:*

$$f(X_{m-k}, \ldots, X_m) = f(X_\tau) \tag{7}$$

$$f(X_{m-k}, \ldots, X_m, X_i) = f(X_i) \tag{8}$$

*where $X_{m-k}, \ldots, X_m$ are parent nodes of node $X_i$; $f(X_\tau)$ is represented as the probability model of the last occurrence in all parent nodes; and $f(X_i)$ is the probability model of node $X_i$, which conforms to the regular pattern of discrete time.*

**Proof.** The structure of the FGHP Bayesian network indicates that the conditional probability model of node $X_i$ is determined by itself and all parent nodes. Each parent node belongs to different parallel subprocesses of FGHP; thus, the prior probability model is adapted as follows:

$$f(X_{m-k}, \ldots, X_m) = \prod_{v=m-k}^{m} f(X_v) \tag{9}$$

where $f(X_v)$ is the occurring probability model of node $X_v$. Simultaneously, FGHP's Bayesian network evolves without aftereffect whilst a parent node $X_{v'}$ is completed, and the prior probability model is optimized as follows:

$$f(X_{m-k}, \ldots, X_m) = \frac{\prod\limits_{v=m-k}^{m} f(X_v)}{f(X_{v'})} \tag{10}$$

where $f(X_{v'})$ is the probability model of node $X_{v'}$. According to the elimination method and network structure constraints, assuming that node $X_{v'}$ is the final parent node of $X_i$, the prior probability model of all parent nodes is the same as that of node $X_{v'}$, and the joint probability model is that of node $X_i$. Eventually, Theorem 1 is proven to provide the regulation of time prediction of the FGHP node. □

As the flight operation flow advances, the configuration of the Bayesian network model for FGHP changes, and the prior probability model of the corresponding subprocess is revised synchronously. Combining the attributes of flight operations and the Bayesian network model of the FGHP, a dynamic time prediction of the FGHP node algorithm (Algorithm 1) is proposed based on the evolution of the FGHP node, the specific steps of which are demonstrated. Adaptive kernel probability density estimation is selected to update the prior and joint probability models of FGHP nodes.

---

**Algorithm 1:** Time prediction of FGHP nodes

---

**Input:** Historical sample space $\Omega$, total number of samples $N$, attribute set of pending FGHP node time prediction $S^0$
**Output:** FGHP node time prediction $X_i T'$
1  Initialize the sample space $\Omega_0$ and the number of samples $n_0$ for the predicting FGHP;
2  $j = 0, n_0 = 0$;
3  **While** $j \leq N$ **do**
4    $S^j \leftarrow (\xi(j), \Omega)$ ; /*assign the attribute of sample $\xi(j)$ in $\Omega$ to $S^{j}$*/
5    **if** $S^j == S^0$
6      $\Omega_0 \leftarrow \xi(j)$ ;/*select same attributes samples */
7      $n_0 = n_0 + 1$;
8    **end if**
9    **return** $\Omega_0$;
10  **end while** /*generate the probabilistic inference sample space*/
11  **for** $i = 1 : 11$ /*loop for each node*/
12      **for** $h = i : 11$
13        $X'_h \leftarrow (\xi^h, \Omega_0)$  /*extract sample set of each node*/
14        $f(X_h) \leftarrow K(X_h, X_{h'})$  /*updating node probability model*/
15      **end for**
16      Probabilistic reasoning based on Bayesian network of FGHP;
17      $X_i T' \leftarrow \max\{f(X_i | \ldots)\}$  /*maximize the conditional probability node of FGHP as predicted result*/
18      $\mathbf{X}\mathbf{T}' \leftarrow X_i T'$
19  **end for**
20  **return** $\mathbf{X}\mathbf{T}'$

---

## 3. Dynamic Performance Evaluation Methods

In this section, performance evaluation can be interpreted as the degree to which the system's goals are achieved or the system expects a set of specific task requirements.

In addition, FGHP performance evaluation is the measurement of the gap between the current and future state or expectations of the actual process. As the nodes of FGHP are updated with time evolving, some nodes will be converted from non-occurring to having occurred. Changes in node status might cause gaps between FGHP's predictions and expected performance results. Aiming at comprehensively and systematically evaluating the performance of ground handling for all flights in the airport, we introduce the basic structure of multi-FGHP performance evaluation and relevance to A-CDM, which can predict the evolution of FGHP nodes in real time based on shared information. Then, the performance evaluation of a single FGHP is performed in accordance with the result of nodes dynamically. Subsequently, a performance evaluation method system for the multiflight ground handling process is proposed. It can provide an objective decision basis for airport operation and flow pushback control and ground support resource scheduling.

### 3.1. Structure of Performance Evaluation System for Multi-FGHP

In the implementation of A-CDM, information sharing is the primary and significant step. An objective performance evaluation system for FGHP is particularly critical, and it should promote airport operators accurately perceiving the airport operation situation from the complicated information and making reasonable decisions in collaboration with various participants. The structure of the multi-FGHP performance evaluation system is shown in Figure 3. All information of the FGHP in the airspace of the airport is uploaded to the information-sharing platform of A-CDM periodically, and the completed FGHP dataset is stored in the historical database. Combining historical and real-time data, the nodes of FGHP are dynamically predicted using the time prediction algorithm (proposed in Section 2.2). The operation of the airspace must comply with the corresponding standards, which provide a reference for the performance evaluation of FGHP. The past, present and future statuses of each FGHP node are mastered to construct a performance evaluation framework of multi-FGHP. Then, the historical data are used to acquire the FGHP node expectations under different conditions. The corresponding single FGHP performance evaluation results are obtained by the relationship between actual, forecast and expectation. Finally, combined with the superposition principle of multi-FGHP, a performance evaluation system structure is designed on the basis of the process evolution.

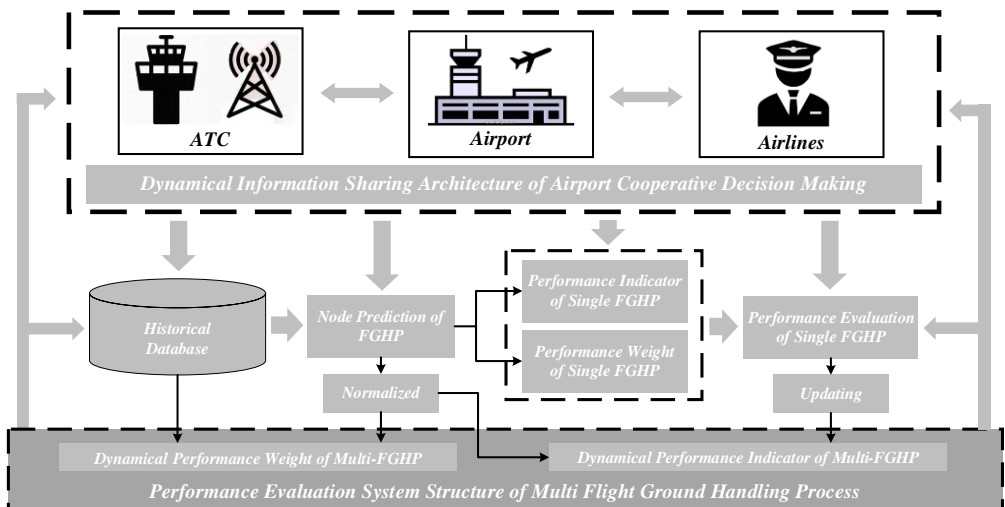

**Figure 3.** Multi-FGHP performance evaluation system structure.

The proposed multi-FGHP performance evaluation method is based on the following assumptions:

i. Flight landings must be taxied to the designated stands strictly in accordance with the assigned path.

ii.     The handover time or waiting time of the same piece of equipment in the same ground handling node is not considered in the performance evaluation method of multi-FGHP.

iii.    Airport ground handling resource allocation, routing and scheduling are idealized.

*3.2. Performance Evaluation of Single FGHP*

The multi-FGHP performance evaluation architecture is manifested in Figure 3. An original single FGHP performance evaluation method is also significant for ensuring access to objective decision-making information. In accordance with node time prediction and real-time completion results combined with each node expectation of a single FGHP, the dynamic performance evaluation includes performance indicators and weight.

**Definition 1.** *If $E(X_iT)$ is the expected time of node $X_i$ and $X_iT$ is its actual time, then the performance indicator of node $X_i$ (represented $PI(X_i)$) can be measured as the absolutely generalized Euclidean distance $ED(X_i)$ between $E(X_iT)$ and $X_iT$ ($X_iT'$ instead if not occurred), as shown in Equations (11) and (12):*

$$ED(X_i) = \sqrt{|E^2(X_iT) - X_iT^2|} \tag{11}$$

$$PI(X_i) = \exp\left\{\frac{ED(X_i)}{\sigma}\right\} \tag{12}$$

*where Equation (12) is the normalized function for the performance indicator of node $X_i$, and $\sigma$ is the reasonable tolerance of the normalized function, obtained from the historical database by the three-sigma rule. Simultaneously,*

$$E(X_iT) = \frac{\sum\limits_{j=1}^{N_i} X_iT^j}{N_i} \tag{13}$$

*where $N_i$ is the number of flights with the same attributes, and $X_iT^j$ is the time value for node $X_i$ of flight j among them.*

**Definition 2.** *If $X_iT$ is the actual time of node $X_i$ and $X_iT'$ is its dynamic prediction time, then the performance weight of node $X_i$ (represented $PW(X_i)$ ) can be calculated in the Cauchy function to describe the importance and adjustability, as shown in Equations (14) and (15)*

$$PW'(X_i) = \begin{cases} 1 & occurred \\ 1 + \frac{1}{1+0.1(X_iT'-E(X_iT))^2} & otherwise \end{cases} \tag{14}$$

$$PW(X_i) = \frac{PW(X_i)}{\sum\limits_{k=1}^{11} PW'(X_k)} \tag{15}$$

*where $PW'(X_i)$ is considered as the performance weight of node $X_i$ before normalization and Equation (15) is mainly applied to normalize the performance weights for all nodes of FGHP.*

Assuming that a flight *j* exists in the ground handling process, a certain number of FGHP time datasets with the same aircraft type, airline and route attributes as flight *j* and no delays are extracted as the historical database to calculate the expected time for each node of FGHP. The time of each unoccupied node is predicted using the proposed algorithm. The performance indicators and weights of flight *j* are dynamically computed

according to the expected, actual and predicted time of each node, and the single FGHP performance evaluation value $SPE(j)$ is dynamically calculated, as shown in Equation (16).

$$SPE(j) = \sum_{i=1}^{11} PI(X_i)PW(X_i) \tag{16}$$

As shown in Equation (16), the performance evaluation $SPE(j)$ of a single FGHP is determined by performance indicators and weights, which cause the evolution of FGHP to promote the update of $SPE(j)$. Evidently, the range of $SPE(j)$ is $[0, 1]$.

*3.3. Multi-FGHP Performance Evaluation Methods*

With the real-time fluctuations in the performance evaluation of a single FGHP, the performance indicator vector **MPI** of multi-FGHP is formed by the performance evaluation results of a single FGHP.

$$\boldsymbol{MPI} = [SPE'(1), SPE'(2), \cdots, SPE'(j), \cdots, SPE'(M(t))] \tag{17}$$

where $SPE'(j)$ is the normalized result of single FGHP performance evaluation value $SPE(j)$ as in Equation (18), and $M(t)$ is the number of flights that stand in the FGHP in real time as well as $D(t)$ departure, which is sorted in **MPI** by time logic.

$$SPE'(j) = \frac{D(t)SPE(j)}{M(t)\sum\limits_{k=1}^{M(t)} SPE(k)} \tag{18}$$

According to the propagation effects of FGHP, the interaction of multi-FGHP is an increasing–stable–decreasing process. It is also related to the nearest status of FGHP (inbound handling, ground handling and departure handling) in the airport. Assuming that the actual arrival time of flight $j$ is $X_0 T^j$, the multi-FGHP performance weight function $W(j)$ is presented with prior parameters.

$$W'(j) = \begin{cases} 1 / \left\{ 1 + \exp\left\{ -\alpha_1 [(X_i T^j - X_0 T^j) - \beta_1] \right\} \right\} & i = 1, 2 \\ 1 & i = 3, 4, 5, 6, 7, 8, 9 \\ \exp\left\{ -\alpha_2 [(X_i T^j - X_0 T^j) - \beta_2]^2 \right\} & i = 10, 11 \end{cases} \tag{19}$$

$$W(j) = \frac{W'(j)}{\sum\limits_{k=1}^{M(t)} W(k)} \tag{20}$$

where $W'(j)$ is the multi-FGHP unnormalized performance weight for flight $j$. The prior parameters, including $\alpha_1$, $\alpha_2$, $\beta_1$ and $\beta_2$, are related to the aircraft type of FGHP, which can be obtained from the analysis and mining of the actual operational support data of the airport. Combined with the actual operating standard, the settings of prior parameters for domestic mainstream aircraft are shown in Table 1.

**Table 1.** Prior parameters for domestic mainstream aircraft in terms of the evolution of the ground handling process for different aircraft types.

| Parameters | A320 Series | B737 Series |
|:---:|:---:|:---:|
| $\alpha_1$ | 0.1952 | 0.1876 |
| $\alpha_2$ | 0.0056 | 0.0059 |
| $\beta_1$ | 23.9430 | 25.7765 |
| $\beta_2$ | 25.9954 | 27.3665 |

For adaptation to the multi-FGHP performance indicator vector, the performance weights $W(j)$ are transformed into diagonal matrix **MPW**, as shown in Equation (21). In

the equation, the coupling relationship of successive multi-FGHP is clarified to some extent. The multi-FGHP performance evaluation method is proposed to consist of a matrix of performance indicator vectors and weights, where the area formed by the indicators and weights of a single FGHP is the core of the method, and the multi-FGHP performance evaluation is defined.

$$
MPW = \begin{bmatrix}
\frac{W(1)+W(2)}{2} & 0 & \cdots & 0 & \cdots & 0 & 0 \\
0 & \frac{W(2)+W(3)}{2} & \cdots & 0 & \cdots & 0 & 0 \\
\cdots & \cdots & \ddots & 0 & \cdots & \cdots & \cdots \\
0 & \cdots & 0 & \frac{W(j)+W(j+1)}{2} & 0 & \cdots & 0 \\
\cdots & \cdots & \cdots & 0 & \ddots & 0 & \cdots \\
0 & 0 & \cdots & \cdots & 0 & \frac{W(M(t-1))+W(M(t))}{2} & 0 \\
0 & 0 & \cdots & 0 & \cdots & 0 & \frac{W(M(t))+W(1)}{2}
\end{bmatrix}_{M(t)\times M(t)} \tag{21}
$$

**Definition 3.** *Assuming that* **MPI** *is updated by single FGHP performance evaluation results, the multi-FGHP expected performance indicator vector* **EMPI** *is constructed by an $M(t)$-dimensional dynamical vector, where all elements of* **EMPI** *are the maximum upper limit of a single FGHP performance evaluation value (shown in Equation (22)).* **MPI** *and* **EMPI** *are instantly calculated by sharing data of the A-CDM system deployed in AOCC. Combined with the advanced transformation of performance weights, as in Equation (23), the multi-FGHP performance evaluation MPE is the ratio of the area of the performance indicator of the adjacent FGHP AS to the expected area ES. The detailed calculation process is derived using Equation (24)*

$$
\textbf{EMPI} = [1, 1, \cdots, 1]_{1 \times M(t)} \tag{22}
$$

$$
\textbf{MPW}^{'} = \textbf{sin}(2\beta * \textbf{MPW}) \tag{23}
$$

$$
\begin{cases}
AS = \textbf{MPI} * \textbf{MPW}^{'} * \textbf{MPI}^{'\mathrm{T}} \\
ES = \textbf{EMPI} * \textbf{MPW}^{'} * \textbf{EMPI}^{\mathrm{T}} \\
MPE = AS/ES
\end{cases} \tag{24}
$$

*where* **MPI**$^{'}$ *is an elementary transformation of the performance indicator vector* **MPI** *of multi-FGHP, which moves the first element to the end (as shown in Equation (25)).*

$$
\textbf{MPI}^{'} = [SPE'(2), SPE'(3), \cdots, SPE'(M(t)), SPE'(1)] \tag{25}
$$

## 4. Experimental Results

### 4.1. Dataset and Settings

In this section, the proposed FGHP node time prediction algorithm is tested initially using a dataset with 2192 FGHP landings and take-offs from a domestic hub airport in June 2019. A sample of attribute set $S^0$ for FGHP is presented in Table 2, including the aircraft type, boarding gates, nature of airline and density of flight. The actual time for every node of FGHP is shown in Table 3; it is converted into the length of time by subtracting the actual arrival of the flight for each node of FGHP.

The original dataset provides the scheduled time of the aircraft arriving at the off-block state. A completed FGHP is not constituted in the departure and arrival flights. Thus, the transit flights are considered in the designed experiment, as are cargo and mail flights. For the convenience and simplicity of the calculation and analysis, the actual ground handling dataset is processed and optimized as follows:

i.　Fields with many vacancies and illogical and duplicate datasets are deleted, and datasets with one or two missing fields are complemented according to the time expectations of FGHP nodes.

ii.　Inbound, ground and outbound handling datasets are linked and merged, and shared flight datasets are integrated according to the assigned airline.

**Table 2.** Sample of ground handling process attributes, where 15 min is the unit time for sorting of prologue and backorder.

| Elements for $S^0$ | Attributes |
|---|---|
| flight number | MU2836 |
| date | 1 June 2019 |
| aircraft type | A320 |
| nature of airline | domestic short-haul routes |
| boarding | 244 |
| sorting of inbound | 6 |
| sorting of departure | 5 |

**Table 3.** Sample of ground handling process node.

| Node | Actual Time | Converted Result (min) |
|---|---|---|
| $X_0$ | 11:28 | 0.00 |
| $X_1$ | 11:37 | 9.00 |
| $X_2$ | 11:44 | 16.00 |
| $X_3$ | 11:56 | 28.00 |
| $X_4$ | 11:58 | 30.00 |
| $X_5$ | 11:45 | 17.00 |
| $X_6$ | 11:52 | 24.00 |
| $X_7$ | 12:03 | 35.00 |
| $X_8$ | 11:58 | 30.00 |
| $X_9$ | 11:51 | 23.00 |
| $X_{10}$ | 12:14 | 46.00 |
| $X_{11}$ | 12:20 | 50.00 |

*4.2. Results*

The multi-FGHP performance evaluation system structure is supported by the single FGHP performance evaluation method, in which the correlation to FHGP is considered. The proposed performance evaluation theory applied for airport operation situation awareness is dynamic. Thus, the results are demonstrated by complex calculations at a certain moment. The time of each node is predicted immediately to update the single FGHP performance indicators and weights in combination with the actual and expected situation, which is proposed in Sections 2.2 and 3.2. Each flight in the ground processing status is evaluated to obtain multi-FHGP performance evaluation results in real time; this may provide decision information for ground handling of airport operation control.

Suppose that the current time is 12:00 on 1 June 2019; node $X_6$ of FGHP is completed and the domestic transit flight 2836 is assigned by Eastern Airlines boarding in 244.

Figure 4 illustrates that the estimated off-block (node $X_{11}$) time (EOBT) is predicted by the proposed algorithm by determining the maximum value of the node's conditional probability. Similarly, all nodes of ground handling are predicted with the evolution of the process, as shown in Figure 5. When the node has occurred, the prediction result is replaced by the actual value in Figure 5. and the accuracy is gradually improved to 87.63%.

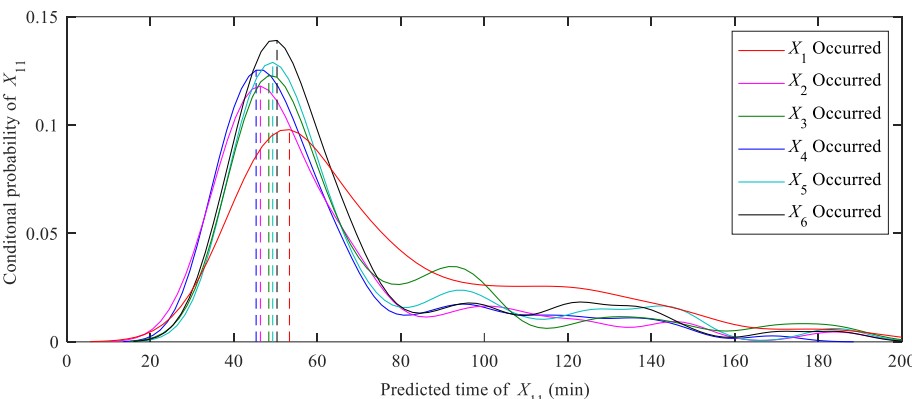

**Figure 4.** Estimated off-block time of FGHP with node updating from $X_1$ to $X_6$).

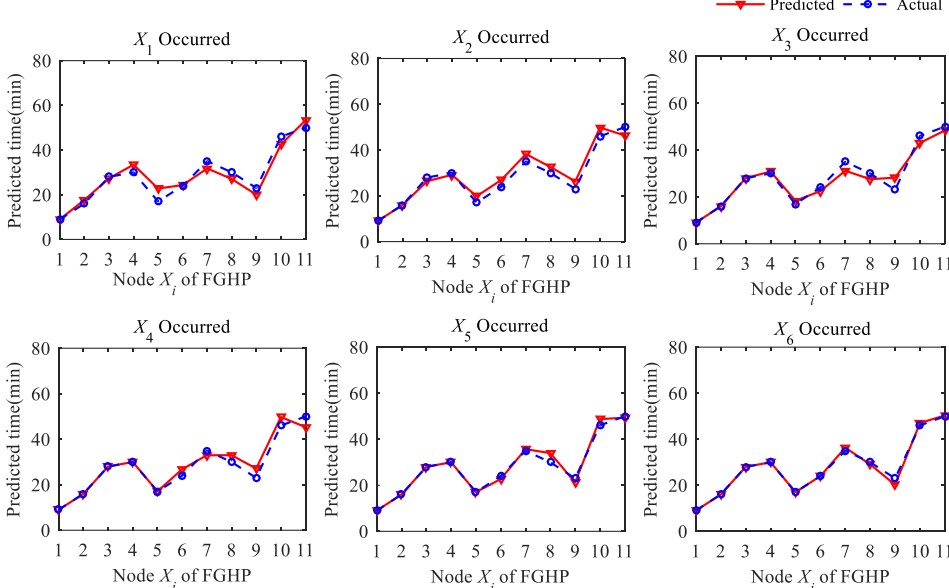

**Figure 5.** Single FGHP node time prediction by the proposed algorithm.

As mentioned in Section 3.2, the performance indicators for the single FGHP calculated by the ground handling process for the A320 series are shorter than those of the B737 series, and gate boarding saves more time than remote boarding. Evidently, the same regular pattern also exists in the international and domestic short-haul/long-haul routes, which might be required with data mining.

Figure 6 shows the single FGHP performance evaluation process for MU2863 combined with the prediction and desired time of FGHP nodes from on-block ($X_1$) to the current moment (as shown in Figure 5 and Table 4. The figure shows that the gap between the actual and expected performance indicators is vividly manifested by the radar chart. The actual performance of this flight (MU2863) gradually deteriorates from $X_1$ to $X_6$, indicating that the ground handling process between the inbound and ground handling process is unbalanced. Table 5 calculates the updated *SPE* of the six nodes, where the value decreases to 0.6979 (as shown in Table 5). The updating of $X_6$ gradually approaches the baseline, thereby demonstrating the viewpoint. Meanwhile, Figure 6 shows that the performance of passenger boarding ending ($X_{10}$) and off block ($X_{11}$) is maintained at a poor level, probably due to the deviation of the ground handling process. Therefore, AOCC can intervene in the ground handling process of MU2863 based on the single FGHP performance evaluation result to avoid delay waves.

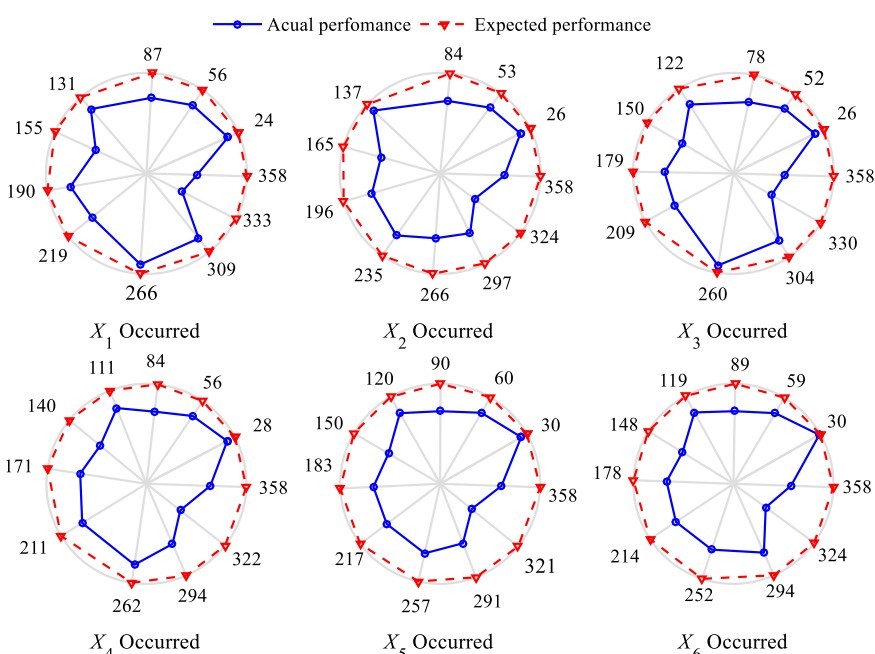

**Figure 6.** Single FGHP performance evaluation process with node updating (from $X_1$ to $X_6$), manifested as radar charts for performance indicators and weights.

**Table 4.** Expectation (min) of performance indicators for single FGHP according to the attribute classification.

| Attributes | $X_1$ | $X_2$ | $X_3$ | $X_4$ | $X_5$ | $X_6$ | $X_7$ | $X_8$ | $X_9$ | $X_{10}$ | $X_{11}$ |
|---|---|---|---|---|---|---|---|---|---|---|---|
| A1 | 7.62 | 21.36 | 31.48 | 41.25 | 35.54 | 79.93 | 88.82 | 90.13 | 47.80 | 103.72 | 105.02 |
| A2 | 6.03 | 13.38 | 22.67 | 26.01 | 15.57 | 26.34 | 30.35 | 31.39 | 25.35 | 50.10 | 54.54 |
| A3 | 5.31 | 14.17 | 24.20 | 26.81 | 18.33 | 31.40 | 35.26 | 36.62 | 26.09 | 50.84 | 56.07 |
| A4 | 6.92 | 16.79 | 31.63 | 38.26 | 39.13 | 80.71 | 84.67 | 87.21 | 43.21 | 102.17 | 109.67 |
| A5 | 6.36 | 14.84 | 25.79 | 27.46 | 30.69 | 28.64 | 43.71 | 47.97 | 26.14 | 66.16 | 70.56 |
| A6 | 6.08 | 14.86 | 25.50 | 27.71 | 21.78 | 30.25 | 36.91 | 38.51 | 26.50 | 55.94 | 61.00 |
| A7 | 8.87 | 17.10 | 46.65 | 52.78 | 49.73 | 91.13 | 91.57 | 93.31 | 54.08 | 110.14 | 120.39 |
| A8 | 5.78 | 32.99 | 38.42 | 38.47 | 40.97 | 45.09 | 53.14 | 59.11 | 36.11 | 73.46 | 77.79 |
| A9 | 8.93 | 32.33 | 45.33 | 63.67 | 50.33 | 50.86 | 77.33 | 77.03 | 41.33 | 96.37 | 103.33 |
| A10 | 7.82 | 24.85 | 51.83 | 45.19 | 56.70 | 90.97 | 101.26 | 103.61 | 64.56 | 123.47 | 127.99 |
| A11 | 8.67 | 16.54 | 27.83 | 33.67 | 74.33 | 43.29 | 88.33 | 91.33 | 30.36 | 110.88 | 120.83 |
| A12 | 6.79 | 15.43 | 32.07 | 32.64 | 27.93 | 55.21 | 50.36 | 56.36 | 64.86 | 76.93 | 88.57 |

Definition of A1: A320 series (A320s), gate boarding (gate-b), international transit flight (t-flight); A2: A320s, gate-b, domestic short-haul (domestic-s-h) t-flight; A3: A320s, gate-b, domestic long-haul (domestic-l-h) t-flight; A4: A320s, remote boarding (remote-b), international t-flight; A5: A320s, remote-b, domestic-s-h t-flight; A6: A320s, remote-b, international t-flight; A7: B737 series (B737s), gate-b, international t-flight; A8: B737s, gate-b, domestic-s-h t-flight; A9: B737s, gate-b, domestic-l-h t-flight; A10: B737s, remote-b, international t-flight; A11: B737s, remote-b, domestic-s-h t-flight; A12: B737s, remote-b, international t-flight.

**Table 5.** Single FGHP performance evaluation value with node updating (from $X_1$ to $X_6$).

| Evolution of FGHP | $X_1$ Occurred | $X_2$ Occurred | $X_3$ Occurred | $X_4$ Occurred | $X_5$ Occurred | $X_6$ Occurred |
|---|---|---|---|---|---|---|
| *SPE* of MU2836 | 0.7472 | 0.7187 | 0.7347 | 0.7132 | 0.6844 | 0.6979 |

According to the proposed multi-FGHP performance evaluation method in Section 3.3, five transit flights (3U8953, HU7335, MU2836, MU2124 and FM9327) are in the ground handling process and initially identified at the current moment. The single FGHP performance indicators and weights of each flight are demonstrated in Figure 7 by the evaluation

approach mentioned in Section 3.2. *SPE* is calculated, as shown in Table 6. Combined with the dynamic value of *SPE* and the process evolution of every flight, the performance indicator vector and weight matrix is constructed and updated. Thus, the final multi-FGHP performance evaluation is presented in the last subfigure of Figure 7. At the same time, the performance evaluation value of multi-FGHP is 0.6651, as shown in Equation (24). Thus, the current overall situation of the airport ground handling process is above the baseline. The range of performance evaluation values is 0–1, and the baseline is set to 0.5.

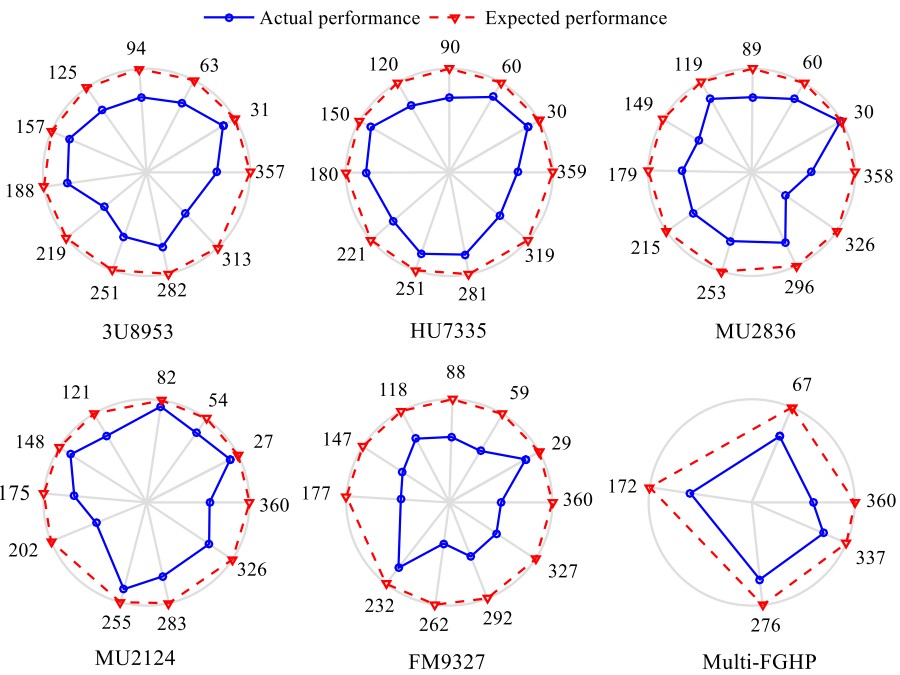

**Figure 7.** Multi-FGHP performance evaluation exhibition with radar charts at the current moment.

**Table 6.** Multi-FGHP performance evaluation value for different flights at the current moment.

| Flight | 3U8953 | HU7335 | MU2836 | MU2124 | FM9327 |
|--------|--------|--------|--------|--------|--------|
| *SPE* for multi-FGHP | 0.6954 | 0.7582 | 0.6979 | 0.7235 | 0.5146 |

The fluctuation of multi-FGHP performance evaluation value is manifested in Figure 8 that is, the initial time is 12:00 and will be updated every 15 min until 18:00. Figure 8 shows that the value of *MPE* changes dynamically over time; it represents the actual status and performance of airport handling management to some extent. In addition, seven sampling points are found below the baseline during the performance evaluation period; they are called singularities. According to the analysis of actual shared data in the A-CDM system, the average punctuality *AP* of airport is calculated in Equations (26) and (27), as follows:

$$s_i(t) = \begin{cases} 0 & 0 < |at_i - pt_i| \leq 15 \\ 1 & |at_i - pt_i| > 15 \end{cases} \tag{26}$$

$$AP = 1 - \frac{\sum\limits_{i=1}^{Q(t)} s_i(t)[(at_i - pt_i) - 15]}{15Q(t)} \tag{27}$$

where $s_i(t)$ is the discriminant function of a delayed flight to be measured by the absolute value of the difference between actual arrival time $at_i$ and scheduled arrival time $pt_i$ for the flight $i$, and $Q(t)$ is the sum of departure flights $D(t)$, ground handling flights $M(t)$

and inbound flights $I(t)$ at the current time. The operating status of each singular point is illustrated in Table 7, where all singularities are flight delays resulting in a decline in the average punctuality rate.

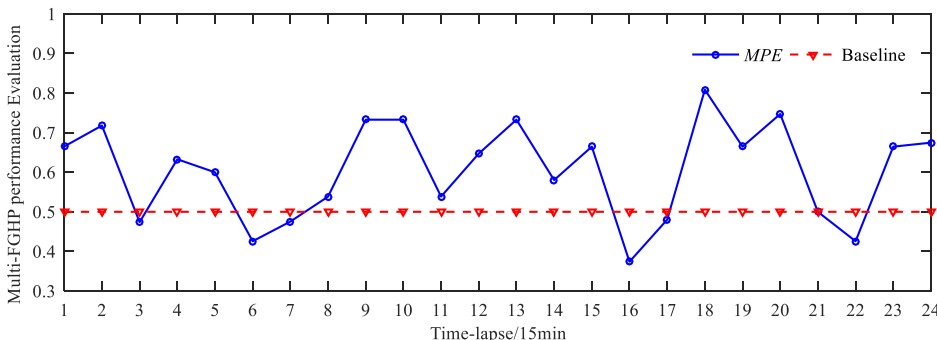

**Figure 8.** Multi-FGHP performance evaluation fluctuation with time lapse.

**Table 7.** Singularity analysis of multi-FGHP performance evaluation at a fixed time (12:00–18:00 on 1 June 2019).

| Singularity | MPE | Delayed/Total | Average Punctuality |
|:---:|:---:|:---:|:---:|
| 3 | 0.4738 | 1/5 | 0.8933 |
| 6 | 0.4256 | 1/7 | 0.8476 |
| 7 | 0.4766 | 1/6 | 0.8861 |
| 16 | 0.3721 | 2/6 | 0.8111 |
| 17 | 0.4801 | 1/4 | 0.8913 |
| 21 | 0.4998 | 0/6 | 1 |
| 22 | 0.4110 | 1/8 | 0.8833 |

For the existing performance evaluation of airport operation and turnaround process mentioned in Section 1, no real-time performance evaluation methods are available to perceive the objective environment and status of the airport. Thus, a comparative analysis is difficult to carry out. Consequently, the entropy method is designed as a weight update method for performance evaluation of multi-FGHP, and all singularities are evaluated by the modified method. The different performance evaluation values are demonstrated in Figure 9. The proposed performance evaluation method can better reflect the actual situation of the airport ground handling process (average punctuality) than the average punctuality trend. Thus, it is suitable for integrating operational shared information of an airport in the A-CDM system.

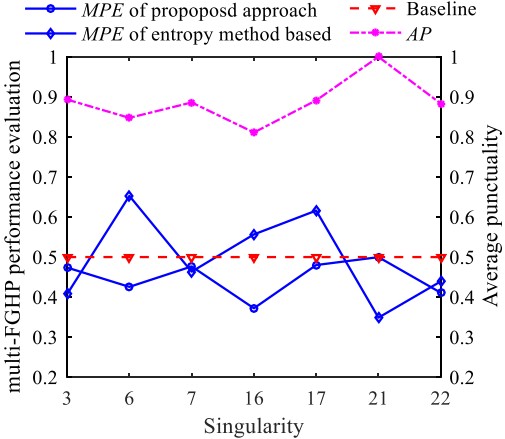

**Figure 9.** Illustration of performance evaluation for singularity by different approaches.

## 5. Conclusions

To perceive the ground handling process of multiple flights to guide airport operations and launch control, the nodes of the ground handling process are predicted dynamically based on the conditional probability reasoning method, which can be regarded as input to the single-FGHP performance evaluation model. A multi-FGHP performance evaluation model is established by the underlying performance indicator and weight matrix which can be calculated by updated shared information and a historical database. The multi-FGHP performance evaluation results can obtain the real-time operation status of the airport from the huge and complex information, which can relieve the pressure on decision-makers analyzing the situation of the scene.

The results also indicate that the comprehensive punctuality rate of flights is affected by the performance of the ground handling process, which is determined by the update of the EOBT in the A-CDM system. The performance evaluation results may function in guiding the airspace operation control to improve the overall operation efficiency of the airport. However, to achieve situational awareness, other factors need to be considered at the same time, such as the safety and capacity of runways and taxiways, the allocation of ground handling resources, air traffic flow control and operation under adverse conditions. Moreover, A-CDM has been promoted to integrate transportation, called A-CDM plus, of which automatic decision-making based on massive shared data is an important part. Evidently, objective performance evaluation may perceive effective information from the data by following the proposed approach. In addition, the application of the proposed performance evaluation method to other scenarios in airport operation, such as terminal collaborative operation and intelligent service process, is worthy of investigation.

**Author Contributions:** Conceptualization, B.L., L.W. and Z.X.; methodology, B.L. and Z.X.; software, B.L. and Q.L.; formal analysis, B.L. and L.W.; validation, B.L. and Z.X.; supervision, B.L.; data curation, Q.L.; writing—original draft preparation, B.L. and Z.X.; writing—review and editing, L.W. and Q.L.; funding acquisition, Z.X. All authors have read and agreed to the published version of the manuscript.

**Funding:** This research was funded by National Key Research & Development Program of China grant number 2018YFB1601200 and the Fundamental Research Funds for the Central Universities grant number 3122019094.

**Institutional Review Board Statement:** Not applicable.

**Informed Consent Statement:** Not applicable.

**Data Availability Statement:** Not applicable.

**Conflicts of Interest:** The authors declare no conflict of interest.

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
