# Peer review of "Performance Evaluation of Multiflight Ground Handling Process"

_aerospace, doi:10.3390/aerospace9050273_

Round 1
Reviewer 1 Report
The presented article deals with the efficiency of ground handling. Given the increasing number of flights, this is a topic addressed by many airports, and I therefore consider the focus of the article to be topical. The article creates a dynamic model for evaluating the performance of ground handling.
The introductory part lacks the authors a sufficient analysis of previous studies in the field. However, the authors explain this shortcoming in the final part of section 4.2. Results where they state: "For the existing performance evaluation of airport operation and turnaround process mentioned in Section 1, no real-time performance evaluation methods are available to perceive the objective environment and status of the airport. Thus, comparative analysis is difficult to carry out.“
The study is written clearly and contains all the necessary parts. The individual sections are listed clearly and distinctly. The text contains many formulas, which are sufficiently explained, thats why the text is easy to read and understand - all formulas and parameters of the research are clearly stated and explained.
The results are clearly written and the proposed theory is tested by real values.
The conclusion of the article is formulated as a discussion. Although brief, it contains summary, the research limits formulated by the authors and other elements by which the research needs to be extended.
In my opinion, the article is suitable for publication in the form in which it was registered. I have no major suggestions for improving or supplementing this article.
Author Response
Dear Aerospace Reviewer,
Thank you for your positive comments concerning our manuscript entitled ‘‘Performance evaluation of Multiflight ground handling process’’ (ID: aerospace -1696525). Those comments are all valuable and very helpful for improving our manuscript, as well as the important guiding significance to our researches. We have studied comments carefully and have made correction which we hope meet with approval. On behalf of my co-authors, we would like to express our great appreciation for your kind work and consideration on publication of our paper.
Thank you and best regards.
Yours sincerely,
Dr. Biao Li
College of Aeronautical Engineering
Civil Aviation University of China
No. 2898, Jinbei Road, Dong li District
Tianjin, China

Reviewer 2 Report
The current paper proposes a dynamic performance evaluation method for multi-FGHP process that the model is also applied for a case study. The paper is well-written, discussing an interesting topic which can be helpful for the related sector. There are some minor comment left:
1- The NP-hard should be defined.
2- A conclusion of the literature review is needed to show the research gap and how your study addresses this gap.
3- The legend is needed for Fig1.
4- It should be clear why i=2,7, 8 in equation 1 (and for other equations).
Author Response
May 14, 2022
Dear Aerospace Reviewer,
Thank you for comments concerning our manuscript entitled ‘‘Performance evaluation of Multiflight ground handling process’’ (ID: aerospace -1696525). Those comments are all valuable and very helpful for revising and improving our manuscript, as well as the important guiding significance to our researches. We have studied comments carefully and have made correction which we hope meet with approval. Revised portion are marked in yellow in the paper, which has been submitted to the academic editor for pending decision. The main corrections in the paper and the responds to your comments are as flowing.
On behalf of my co-authors, we would like to express our great appreciation for your kind work and consideration on publication of our paper.
Thank you and best regards.
Yours sincerely,
Dr. Biao Li
College of Aeronautical Engineering
Civil Aviation University of China
No. 2898, Jinbei Road, Dong li District
Tianjin, China
